# A Missense Variant in *ALDH5A1* Associated with Canine Succinic Semialdehyde Dehydrogenase Deficiency (SSADHD) in the Saluki Dog

**DOI:** 10.3390/genes11091033

**Published:** 2020-09-02

**Authors:** Karen M. Vernau, Eduard Struys, Anna Letko, Kevin D. Woolard, Miriam Aguilar, Emily A. Brown, Derek D. Cissell, Peter J. Dickinson, G. Diane Shelton, Michael R. Broome, K. Michael Gibson, Phillip L. Pearl, Florian König, Thomas J. Van Winkle, Dennis O’Brien, B. Roos, Kaspar Matiasek, Vidhya Jagannathan, Cord Drögemüller, Tamer A. Mansour, C. Titus Brown, Danika L. Bannasch

**Affiliations:** 1Department of Surgical and Radiological Sciences, University of California Davis, Davis, CA 95616, USA; ddcissell@ucdavis.edu (D.D.C.); pjdickinson@ucdavis.edu (P.J.D.); 2Department of Clinical Chemistry, VU University Medical Center, 1081 HV Amsterdam, The Netherlands; E.Struys@vumc.nl (E.S.); b.roos@amsterdamumc.nl (B.R.); 3Institute of Genetics, Vetsuisse Faculty, University of Bern, 3001 Bern, Switzerland; anna.letko@vetsuisse.unibe.ch (A.L.); vidhya.jagannathan@vetsuisse.unibe.ch (V.J.); cord.droegemueller@vetsuisse.unibe.ch (C.D.); 4Department of Pathology, Microbiology and Immunology, University of California Davis, Davis, CA 95616, USA; kdwoolard@ucdavis.edu; 5Department of Population Health and Reproduction, University of California Davis, Davis, CA 95616, USA; miraguilar@ucdavis.edu (M.A.); eabrown@ucdavis.edu (E.A.B.); drtamermansour@gmail.com (T.A.M.); ctbrown@ucdavis.edu (C.T.B.); 6Department of Pathology, University of California San Diego, La Jolla, CA 92093, USA; gshelton@health.ucsd.edu; 7Advanced Veterinary Medical Imaging, Tustin, CA 92780, USA; mbroome@avmi.net; 8College of Pharmacy and Pharmaceutical Sciences, Washington State University, Spokane, WA 99202, USA; mike.gibson@wsu.edu; 9Harvard Medical School, Boston, MA 02115, USA; Phillip.Pearl@childrens.harvard.edu; 10Fachtierarzt fur Kleintiere, Am Berggewann 13, 65199 Wiesbaden, Germany; fk@neurovet.de; 11Department of Pathobiology, School of Veterinary Medicine, University of Pennsylvania, Philadelphia, PA 19104, USA; tomvw@vet.upenn.edu; 12College of Veterinary Medicine, University of Missouri, Columbia, MO 65211, USA; ObrienD@missouri.edu; 13Clinical and Comparative Neuropathology, Ludwig-Maximilians-Universitaet München, 80539 Munchen, Germany; kaspar.matiasek@neuropathologie.de; 14Department of Clinical Pathology, School of Medicine, Mansoura University, Mansoura 35516, Egypt

**Keywords:** inborn error of metabolism, encephalopathy, SSADHD, *ALDH5A1*, GABA, 4-hydroxybutyric acid, succinic semialdehyde, encephalopathy, whole-genome sequencing, precision medicine, GWAS, inherited

## Abstract

Dogs provide highly valuable models of human disease due to the similarity in phenotype presentation and the ease of genetic analysis. Seven Saluki puppies were investigated for neurological abnormalities including seizures and altered behavior. Magnetic resonance imaging showed a diffuse, marked reduction in cerebral cortical thickness, and symmetrical T2 hyperintensity in specific brain regions. Cerebral cortical atrophy with vacuolation (status spongiosus) was noted on necropsy. Genome-wide association study of 7 affected and 28 normal Salukis revealed a genome-wide significantly associated region on CFA 35. Whole-genome sequencing of three confirmed cases from three different litters revealed a homozygous missense variant within the aldehyde dehydrogenase 5 family member A1 (*ALDH5A1*) gene (XM_014110599.2: c.866G>A; XP_013966074.2: p.(Gly288Asp). *ALDH5A1* encodes a succinic semialdehyde dehydrogenase (SSADH) enzyme critical in the gamma-aminobutyric acid neurotransmitter (GABA) metabolic pathway. Metabolic screening of affected dogs showed markedly elevated gamma-hydroxybutyric acid in serum, cerebrospinal fluid (CSF) and brain, and elevated succinate semialdehyde in urine, CSF and brain. SSADH activity in the brain of affected dogs was low. Affected Saluki dogs had striking similarities to SSADH deficiency in humans although hydroxybutyric aciduria was absent in affected dogs. *ALDH5A1*-related SSADH deficiency in Salukis provides a unique translational large animal model for the development of novel therapeutic strategies.

## 1. Introduction

Inborn errors of metabolism (IEMs) are a group of diseases caused by an enzymatic deficiency in a metabolic pathway, most commonly caused by a genetic mutation. While individually these diseases are rare, as a group they are relatively common, with more than 500 IEM diseases reported in people [1]; in animals, they are becoming increasingly recognized [2,3,4,5,6]. In diseases caused by an IEM, clinical signs are due to the reduced or lack of production of a biochemical product, or accumulation of an abnormal amount of substrate or substrates produced by alternative metabolic pathways, secondary to the enzymatic deficiency. The diagnosis of an IEM may be a challenging, as clinical signs can be vague and non-specific, and targeted diagnostic testing is required [7]. IEMs are often recognized in young people and animals, and many have neurological manifestations [8,9].

Seizures are a common neurological sign in dogs [10]. Disorders causing seizures arise either extracranially (reactive seizures), or intracranially [11] Epilepsy is a brain disease characterized by a lasting predisposition to generate seizures, which is classified in dogs as structural epilepsy or idiopathic epilepsy (OMIA 000344-9615) [11]. Causes of structural epilepsy include inflammation (e.g., granulomatous meningoencephalitis), neoplasia, nutritional alterations (e.g., thiamine deficiency), infection, anomalous entities (e.g., hydrocephalus), inborn errors of metabolism and trauma [11]. Dogs with idiopathic epilepsy (IE) are typically 6 months to 6 years of age and usually have normal physical and neurological examinations between seizures [12]. Dogs younger than 6 months or older than 6 years of age usually have reactive seizures or structural epilepsy, rather than idiopathic epilepsy [12].

A seizure disorder reported in Salukis is called central nervous system status spongiosus in Saluki dogs (SSSD). There are only brief reports of this disease in the literature [13,14]. One affected 8-month-old male puppy from a litter of 9 was reported with a 5 month history of seizures and behavioral changes. The sire and dam were full siblings. All nine puppies and the sire were euthanized; pathological changes were noted in the affected puppy and in two clinically normal littermates, and the rest of the puppies and the sire were pathologically normal. Pathological changes in the clinically affected puppy included widespread bilaterally symmetrical status spongiosis of the cerebrum, brainstem and cerebellum at the grey–white matter junction, which extended into both the grey and white matter. There were also lesions in the thalamus, optic nerve and internal capsule but no lesions were noted in the spinal cord [13].

Recognized causes of early-onset symmetrical brain lesions include metabolic, nutritional and toxin-induced diseases [15]. In Saluki dogs with SSSD, the clinical signs and lesions on MRI and pathology appear to be breed specific, identical in distribution and type, and diagnosed in multiple dogs over a long period of time (1987 [13] to 2020). Clinical signs developed while puppies were still with the breeders, making toxicity a less likely cause; pathology differed from previously reported nutritional [16] or toxic [17,18,19] central nervous system problems, and thus a genetic cause was considered most likely. Although a metabolic disorder was not identified by routine diagnostic testing in affected Salukis, an underlying genetic abnormality causing a metabolic problem was most likely based on the age of onset of clinical signs.

The purpose of this study was to define the phenotype of Salukis with SSSD and to determine the underlying genetic cause in this breed. Comprehensive evaluations including MRI and necropsy, as well as metabolic and enzyme activity testing, were performed on urine, serum, cerebrospinal fluid and brain tissue from four affected puppies from two litters from the USA and a litter with three affected puppies from Germany. All seven affected dogs were used for a genome-wide association study (GWAS) followed by whole-genome sequence analysis of three affected puppies, which identified a private homozygous missense variant in the canine *ALDH5A1* gene.

## 2. Materials and Methods 

### 2.1. Affected Dogs

From 2005 to 2015, seven Saluki dogs affected with SSSD from the USA (4) and Germany (3) had DNA collected. Four dogs were examined—three dogs at the William R. Pritchard Veterinary Medical Teaching Hospital at the University of California Davis (UCD) and one dog was evaluated at Fachtierarzt fur Kleintiere, in Wiesbaden, Germany. All 4 dogs were presented by their breeders for examination. A fifth Saluki dog affected with SSSD from the USA had a necropsy completed at UCD Two additional German dogs were not evaluated clinically beyond the breeder’s description of the clinical signs.

### 2.2. Control Dogs

#### 2.2.1. MRI Evaluation

Four unaffected Saluki dogs related to the affected USA Saluki dogs were examined and had magnetic resonance imaging of their brain and completed at Advanced Veterinary Medicine Imaging in Los Angeles, California. 

#### 2.2.2. Targeted Metabolic Testing

Archived urine = 4, serum = 4, cerebrospinal fluid (3) and brain tissue (4) from 15 different non-Saluki dogs unaffected by SSSD were utilized as control samples. 

### 2.3. Affected Saluki Dogs

Blood work (complete blood count, and serum biochemical profile) was performed by the referring veterinarians in two dogs (dogs 3 and 4). Cerebrospinal fluid sample (CSF) was collected in two dogs (dogs 1 and 5). One CSF sample was routinely analyzed in Germany and the other sample was collected at UCD and frozen at −80 degrees for further analysis. Two dogs had quantitative urine organic acid testing completed at the University of California San Diego Biochemical Genetics Laboratory (dogs 1 and 2). Urine was shipped to the lab by the breeder and was analyzed by the lab 18 days later. Four dogs had complete necropsies completed at UCD, and one had a necropsy at Ludwig Maximilians Universität München in Germany. The owners consented to the necropsy and processing of postmortem samples. Following necropsy, the brain was immediately immersed in 10% neutral buffered formalin followed by standard paraffin embedding. Selected regions were sectioned at 5 µm slice thickness and stained with hematoxylin-eosin and luxol fast blue-cresyl violet.

#### MRI and Histopathology

Six Saluki dogs underwent MRI of the brain—two affected Saluki dogs underwent magnetic resonance imaging (MRI) of the brain at UC Davis, and four unaffected Saluki dogs had imaging of the brain at Advanced Veterinary Medicine Imaging in Los Angeles, California. Both locations used a 1.5 T MRI system (GE Signa, GE Healthcare, Waukesha, WI, USA), with paired 5″ general purpose radiofrequency coils. Sagittal T1-weighted (T1W) and T2-weighted (T2W) images, transverse T1W, T2W, fluid attenuating inversion recovery (FLAIR), and T2*-weighted (T2*W) images, and dorsal T2W images were acquired of the brain. Sagittal, transverse, and dorsal T1W images were repeated after intravenous administration of 0.1 mmol/kg gadopentetate dimeglumine (Magnevist, Bayer, Whippany, NJ, USA). 

### 2.4. Sample Collection and DNA Extraction

Blood samples, pedigree, and phenotype information were collected from 7 affected dogs and 18 close relatives of affected dogs including 4 parents (Figure 1). Additional samples were collected from 48 healthy Saluki dogs. Healthy dogs of other breeds (*n* = 228) were used that were part of a DNA repository at UC Davis. DNA was extracted from EDTA whole blood samples using Gentra Puregene DNA purification extraction kit (Qiagen, Valencia, CA, USA). Collection of canine samples was approved by the University of California, Davis Animal Care and Use Committee (protocol #18561) and the Cantonal Committee for Animal Experiments (Canton of Bern; permit 75/16).

### 2.5. Genome-Wide Association Scan

SNP genotyping was performed using the Illumina Canine HD 174,000 SNP array (Illumina, San Diego, CA, USA) for 7 affected cases and 28 neurologically normal adult Saluki controls. Genome-wide association analysis was performed using Plink [20]. SNPs were pruned from analysis if the minor allele frequency was <5% and the call rate <90%. Chi-square association analysis, Bonferroni adjustments, and genomic inflation calculations were performed in Plink. Figure 5 was made in R using ggplot2 [21,22].

### 2.6. Whole-Genome Sequence 

Whole-genome sequencing (WGS) was performed on the two affected Salukis from the USA and compared to 98 controls dogs from various breeds as reported, and coverage was 6.4× for 1052 and 5.3× for 5813 SRA: SRR5311685 and SRR5311664 (study: PRJNA377155) [23]. Segregation of variants was performed using 2 cases compared to 98 controls within the homozygous interval identified by visual inspection of genotype calls from the array data. Variant Effect Predictor (VEP), as employed in Ensembl using Refseq annotation and additional EST/CCDs, was used to predict the effect of segregating variants [24]. Polyphen2 [25] and SIFT [26] were used to evaluate the severity of the missense variants.

In one German Saluki (dog 5), whole-genome sequencing using genomic DNA isolated from the blood sample of the affected dog was performed as described previously [27]. Data corresponding to approximately 15× coverage of the genome was collected on an Illumina HiSeq2000 instrument (2 × 100 bp). Read mapping and variant calling were carried out as previously described [28], with respect to the CanFam3.1 genome reference assembly and the NCBI annotation release 105. Variant filtering was performed against 581 dog and 8 wolf genomes which were publicly available [28]. WGS data of the affected dog was made available under study accession PRJEB16012 at the European Nucleotide Archive (www.ebi.ac.uk/ena; sample accession SAMEA4504825). 

Annotations within the canine *ALDH5A1* gene refer to the NCBI mRNA accession no. XM_014110599.2 and the protein accession no. XP_013966074.2. Annotations within the *GPLD1* gene refer to the NCBI mRNA accession no. XM_005640079.3 and the protein accession number XP_005640136.1. Annotations for the putative *PTCHD3* gene refer to the mRNA accession no. ENSCAFT00000039442.3 and the protein accession no. ENSCAFP00000035301. 

### 2.7. Genotyping

The variant in *ALDH5A1* disrupted a Sau96I restriction enzyme site, allowing rapid genotyping by PCR-RFLP analysis. PCR primers were designed using primer 3 [29] to amplify an 872 bp product, which upon digestion with Sau96I produced 700 and 150 bp fragments for the variant allele and 550 and 150 bp fragments for the wild-type allele. All PCR was carried out using Qiagen HotStart DNA polymerase kit (Qiagen, Valencia, CA, USA) at an annealing temperature of 58 degrees using the following PCR primers: F_TCCCGAGTTAGGGGTTCTTT, R_TCACGTTTTCCTGATTTCACC. The same primers were used to verify the mutation by Sanger sequencing on an Applied Biosystems 3500 Genetic Analyzer using the Big Dye Terminator Sequencing Kit (Life Technologies, Burlington, ON, Canada). 

### 2.8. RT-PCR

RT-PCR was performed for liver cDNA from a case and control. *RPS5* was included as a housekeeping gene control [27]. Primers were designed using Primer3 (SAL_2F: TTGTATTTGACAGCGCCAAC, SAL_2R: CAAGGCCAGATTGCTTCAC) except for RPS5, in which the primers were as recommended [30]. Each reaction included 13.9 μL of water, 2 μL of 10× buffer with MgCl_2_, 1 μL of dNTP, 1 μL of each forward and reverse primers (20 μM), 1 μL of HotStarTaq DNA Polymerase (Qiagen, Valenica, CA, USA), and 1 μL of cDNA made from 1000 ng of RNA. Amplified products were visualized on a 2% agarose gel.

### 2.9. Targeted Metabolic Testing


***Gamma-hydroxybutyrate (GHB) and succinate semialdehyde (SSA):***


GHB and SSA in fluids and brain tissue were quantified by isotope dilution mass spectrometric methodology, as previously described [31,32]. 


***The 4,5-dihydroxyhexanoic acid (DHHA):***


Analysis of DHHA in fluids and brain was comparable to that for GHB as previously described [31], with some modifications. ^2^H_3_-DHHA was used as the internal standard and the samples were extracted a single time with ethylacetate. For quantitation, positive chemical ionization was employed.


***Succinic semialdehyde dehydrogenase (SSADH) activity:***


SSADH activity was quantified fluorometrically in brain tissue samples using the NADH/NAD couple and SSA as substrate, as previously described [33].

## 3. Results

### 3.1. Affected Dogs 

Four dogs from two litters from the USA were closely related, and the third litter from Germany was distantly related to the other two litters. There were four females and three males affected (Figure 1). 

### 3.2. Clinical Phenotype

The breeders of affected Saluki puppies noted that puppies were first abnormal between six and ten weeks of age. Historical clinical signs included seizures, abnormal behavior such as episodes of vocalization (Appendix A), and difficulty being aroused from sleep. Four puppies (dogs 1 and 3–5) were evaluated by a board-certified veterinary neurologist (Table 1). No abnormalities were noted on physical examination. On neurological examination, puppies had mild generalized ataxia with thoracic limb hypermetria (two puppies) (Appendix A), absent menace reflex in both eyes and delayed proprioceptive positioning in all four limbs, consistent with a multifocal disease process. Two dogs (dogs 3 and 4) had a normal CBC and serum biochemical profile completed by their referring veterinarian, and two dogs had normal quantitative urine organic acid analysis (Biochemical Genetics Laboratory, University of California San Diego, San Diego, CA, USA). One dog (dog 5) had a normal cisternal cerebrospinal fluid analysis. Five dogs were treated for seizures with oral phenobarbital or levetiracetam. Although clinical signs did not progress, all affected dogs were euthanized as puppies at the request of the breeders when they were still in their care. Puppies were euthanized between three and nine months of age for quality of life concerns, primarily due to the recurrent episodes of vocalization. Five dogs (dogs 1–5) had necropsies completed. 

### 3.3. MRI and Histopathology

Two affected dogs (dogs 1 and 2) and four other related but unaffected dogs had an MRI of the brain. All four unaffected dogs had unremarkable MR images. Both affected dogs exhibited prominent sulci (Figure 2A,C–E) compared to normal dogs (Figure 2F), consistent with diffuse cortical atrophy. Bilateral, symmetrical, T2 and FLAIR hyperintensity was present in the diencephalon, deep cerebellar nuclei (Figure 2A–C), midbrain (Figure 2D), and multiple basal nuclei (Figure 2E). Multifocal, symmetrical T2 and FLAIR hyperintensity was also present in the deep cortical laminae of the grey matter throughout the cerebral cortex (Figure 2A–F). 

No hypointense lesions or signal voids were observed associated within the brain parenchyma on T2*W images. 

Histopathologically, there was severe bilaterally symmetric spongiform change, worse within the mesencephalon (Figure 3), brainstem, and deep cerebellar nuclei, but also severe in the thalamic nuclei and deep cortical grey matter. The corpus striatum was less affected but exhibited similar lesions most notably in the entopeduncular nuclei and putamen. Neurons exhibited single to multiple, clear, well-demarcated vacuoles that compressed and displaced the nucleus (Figure 4). There was marked proliferation of enlarged astrocytes associated with the spongiform change, and both neurons and astrocytes appear affected. The grey matter was more severely affected, particularly at the grey–white matter junction. The spinal cord was not affected. 

### 3.4. Genetic Analysis

Both sexes are affected and in-depth pedigree analysis revealed the presence of a common male ancestor connecting the American and European families (Figure 1). As all parents of affected offspring show no clinical signs, it could be speculated that the observed disease phenotype follows autosomal monogenic recessive inheritance. Genome-wide association was performed using DNA samples from the seven affected dogs (Figure 1) and 28 phenotypically normal Saluki controls. After quality control, there were 108579 SNPs available for association. A single genome-wide significant association signal based on a p_Bonferroni_ (0.006) on CFA 35 (chr35: g23,654,869; p_raw_ 5.27 × 10^−8^) was identified (Figure 5). Furthermore, a 2.683 Mb region of homozygosity was identified in the seven affected dogs on CFA 35: 21,925,974–24,608,949 bp (CanFam3.1).

In order to identify a causative variant, initially paired-end whole-genome sequences of 2 affected puppies from two American litters (1052, 5813: Figure 1) and 98 unaffected controls from various breeds were investigated in the associated interval. There were 35,982 single-nucleotide variants (SNVs) and 16,832 insertion/deletion (indel) variants identified within the critical interval defined by homozygosity: CFA 35: 21,925,974–24,608,949 bp (CanFam3.1). There were 259 SNVs and 41 indels that segregated with the phenotype in the 100 animals. There were three coding variants identified: a synonymous variant (g.22,506,956G>A) in the *GPLD1* gene, and two protein-changing missense variants, g.22,572,768G>A in the *ALDH5A1* gene, and g.23,908,560T>C in the putative *PTCHD3* gene. The synonymous variant in *GPLD1* was not investigated further. 

The two missense variants in *ALDH5A1* (XM_014110599.2: c.866G>A; XP_013966074.2: p.(Gly288Asp)) and *PTCHD3* (ENSCAFT00000039442.3: c.1247T>C; ENSCAFP00000035301: p.(Iso416Thr)) were evaluated to identify whether the substitutions were potentially deleterious. There is a gap in the canine genome assembly that likely contains at least one additional exon of the *ALDH5A1* gene. Aligning the predicted canine ALDH5A1 protein sequence with the human protein sequence places the canine missense variant at amino acid 381 in human (NP_001071.1). Both PolyPhen2 [25] (probably damaging—1.0) and SIFT [26] (deleterious—0.0) predicted this amino acid substitution to be deleterious. It occurs in a well -conserved portion of the protein (Figure 6B). The missense variant in *PTCHD3* is not predicted to affect the protein based on PolyPhen2 [25] (benign-0.436) and SIFT [26] (tolerated-0.05). In addition, based on the known functions of these two proteins and the independent findings presented below, the *ALDH5A1* variant was the only one pursued further.

Independently, the genome of an affected puppy (dog 5) from the German litter was sequenced. No private variants were found in the *GPLD1* and *PTCHD3* genes. Furthermore, only one protein-changing missense variant (*ALDH5A1*: XM_014110599.2: c.866G>A) remained after filtering for homozygous private variants in the region of interest on chromosome 35 against the 589 control genomes from the Dog Biomedical Variant Database Consortium (DBVDC) variant catalog [27].

The *ALDH5A1* variant was confirmed by Sanger sequencing of genomic DNA (Figure 6A). In order to genotype the *ALDH5A1* missense variant, a PCR-RFLP genotyping assay was used (Figure 6C). 

Genotyping of the *ALDH5A1* missense variant was performed in the seven affected dogs used for the GWAS, and the four available parents (Figure 1). The variant was homozygous in all cases and heterozygous in the parents. Siblings and other relatives, as well as unrelated Saluki dogs, were genotyped; 13 were heterozygous carriers and 48 were homozygous wild type. The segregation of the *ALDH5A* variant fits perfectly with the assumed monogenic recessive Mendelian inheritance within the studied family. In-depth pedigree analysis revealed the presence of a common male ancestor connecting the American and European families (Figure 1). To experimentally determine whether the *ALDH5A* variant was a common canine variant, 228 dogs from various other breeds were genotyped and all were found to have the wild-type allele, which was also confirmed by the absence of the variant in 581 dogs from 125 breeds and eight wolves of the DBVDC cohort [27].

The presence of the variant in cDNA of an affected dog was verified by Sanger sequencing of RT-PCR product from liver of an affected Saluki compared to a control unaffected dog. There was no obvious difference in expression level between the case and the control. Quantitative evaluation was not possible since only one affected dog sample was available. Our results have been integrated in the Online Mendelian Inheritance in Animals (OMIA) database (https://omia.org/OMIA002250/9615/).

### 3.5. Targeted Metabolic Testing

Targeted quantitative organic acids were analyzed on urine, serum, CSF, and brain tissue (Table 2) in affected (*n* = 1 to 4) and control dogs (*n* = 2 to 4). Compared to control dogs, there were marked elevations in urine succinate semialdehyde (SSA) and urine 4,5-dihydroxyhexanoic acid (DHHA) but levels of gamma hydroxybutyrate (GHB) in the urine were normal. Serum GHB and serum DHHA from affected dogs were markedly elevated compared to controls. Serum SSA could not be measured in either affected or control dogs. In cisternal cerebrospinal fluid (CSF) and brain, SSA, GHB and DHHA were markedly elevated in the affected dog compared to controls, with the CSF GHB having the highest elevation (by a factor of at least 4800). Activity of succinate semialdehyde dehydrogenase (SSADH) was absent or markedly reduced to 0.18% of normal in the affected dogs compared to control dogs.

## 4. Discussion

A pathogenic variant in the canine *ALDH5A1* gene associated with recessive SSADH deficiency, formerly known as status spongiosus in Saluki dogs (SSSD) [13]. Seven Saluki puppies from two continents had an onset of multifocal cranial neurological signs at 10 weeks of age or younger. Blood work was normal. On MRI of the brain, lesions were similar in all dogs, with bilateral and symmetrical lesions of the cerebrum, brainstem and cerebellum, predominantly affecting grey matter structures. Because this disorder occurred in purebred Saluki puppies from different environments with a consistent clinical phenotype and a normal extracranial work up, structural epilepsy from an inborn error of metabolism was considered the most likely etiology. Using genome-wide association, followed by whole-genome sequencing, a missense mutation in the *ALDH5A1* gene was identified as the presumed cause of status spongiosus in Saluki dogs, previously reported in Saluki puppies [13]. The mutation segregated completely in family members as an apparently fully penetrant monogenic recessive disorder.

The *ALDH5A1* gene encodes the mitochondrial enzyme succinic semialdehyde dehydrogenase (NAD+) (SSADH), which is involved in the catabolism of the inhibitory neurotransmitter gamma-aminobutyric acid (GABA) (Figure 7). GABA is the major inhibitory neurotransmitter in the central nervous system(CNS), where it is utilized in up to 30% of cerebral synapses [34]; it is also found in non-nervous-system tissue. GABA is synthesized from L-glutamate via glutamate decarboxylase (GAD). The first step in GABA metabolism is by GABA-transaminase (GABA-T) to form succinic semialdehyde (SSA). SSA is then oxidized by the mitochondrial protein succinic semialdehyde dehydrogenase (NAD+) (SSADH) to succinic acid, which then enters the tricarboxylic acid (TCA) cycle for energy generation (Figure 7). In people, recognized disorders of GABA synthesis are GAD deficiency, and recognized disorders of GABA degradation are GABA transaminase deficiency and succinic semialdehyde dehydrogenase (NAD+) (SSADH) deficiency.

The SSADH protein is expressed in the mammalian brain, as well as liver, pituitary, heart, ovary and kidney [35]. In people, the *ALDH5A1* gene is located on chromosome 6p22, and is 10 exons long extending over 38 kb [36]. Succinic semialdehyde dehydrogenase (NAD+) is an enzyme which is member of the aldehyde dehydrogenase family of proteins. In people with SSADH enzyme deficiency, SSA is not catabolized to succinic acid, and thus excess levels of SSA build up in tissues and fluids (Figure 7). Excess succinic semialdehyde is converted to 4-hydroxybutyric acid (GHB) by succinic semialdehyde reductase. Excess SSA may also interact with an intermediate in the pyruvate dehydrogenase complex to form 4,5-dihydroxyhexanoic acid (DHHA). People with a deficiency of SSADH have elevations of SSA, DHHA, and GHB in body fluids [35]. The activity of SSADH is reduced in people, and thus levels of SSA rise, with associated high levels of GHB and DHHA (Figure 7) [37]. The hallmark of SSADH deficiency in people is persistent and elevated levels of the GHB in urine, plasma and CSF [37]. The diagnosis is confirmed by molecular genetics by sequencing the *ALDH5A1* gene for pathogenic variants. There is no effective therapy [37].

In Saluki dogs with SSADH deficiency, levels of SSA and DHHA are elevated in urine, serum, CSF and brain, and GHB is elevated in serum, CSF and brain. Unlike in people, where GHB is elevated in urine, the level of GHB in urine in Saluki dogs with SSADH deficiency is normal. Since the activity of succinate semialdehyde dehydrogenase (NAD+) (SSADH) was absent or markedly reduced, along with elevated levels of SSA, DHHA and GHB, we believe that the previously described central nervous system status spongiosus in Saluki dogs should be more appropriately termed succinic semialdehyde dehydrogenase deficiency (SSADHD).

In people, SSADH deficiency is a rare autosomal recessive neurological disorder caused by a mutation in the *ALDH5A1* gene, reported in 1981 (OMIM 271980) [38]. There are 44 unique mutations in the *ALDH5A1* gene, which occur in exons 1–10; there are no other mutations in genes other than *ALDH5A1* associated with SSADH deficiency in people [36]. The clinical features in people include developmental delay, hypotonia, intellectual disability, ataxia, seizures, hyperkinetic behavior, aggression and sleep disturbances. Approximately 50% of patients have seizures, 45% have neuropsychiatric problems such as sleep disturbances, and many patients also have behavioral abnormalities [34], which all worsen with age [39]. The encephalopathy is considered non-progressive and has wide phenotypic heterogeneity from mild to severe. Symptoms are first noted at a mean of 11 months (range 0–44 months of age), with a mean age at diagnosis of 6.6 years (range of 0 to 25 years) [40]. On MR imaging in people with SSADHD, MR images may be normal, or there may be hyperintensities on T2-weighted imaging in the globus pallidus, cerebellar dentate nuclei and brainstem [41]. A small percentage of people are reported with cerebral white matter hyperintensity on MR imaging as well [41]. There is one single case report of the pathology of SSADH deficiency in a young adult, where there was discoloration of the globus pallidi, congestion of the leptomeninges and scar tissue in the cerebral cortex [42].

The pathophysiology of SSADH deficiency in people is complex. The disease is thought to be caused primarily by the elevation of GHB in the brain, particularly during neurodevelopment or due to the imbalance of neurotransmitters [43] but also potentially from oxidative stress in the brain [44]. GHB is a neuromodulator with a wide array of pharmacological effects [45]. It was initially produced as an injectable anesthetic agent but is now longer utilized for this purpose due to adverse effects but is prescribed to treat cataplexy in narcolepsy/cataplexy, opiate dependency and alcoholism, and is a drug of abuse, where its street names include Grievous Bodily Harm, Liquid Ecstasy and Soap [46]. GHB may cause anxiolytic, hypnotic and euphoric effects as well as short-term memory loss and CNS depression, causing sedation [44,47]; intoxication may cause bradycardia, myoclonus and seizures, hypoventilation, coma and death from respiratory depression [33]. GHB is a monocarboxylate that is primarily cleared from the plasma via metabolism in a dose-dependent fashion through the Kreb’s cycle. Renal clearance of GHB is minor and non-linear, due to the carrier-mediated saturable renal reabsorption of GHB through the proximal tubules via sodium-dependent and pH-dependent monocarboxylate transporters. Methods to increase the renal excretion of GHB have been investigated in animals as part of a treatment plan for people with GHB intoxication, such as with the intravenous administration of L-Lactate which increases the renal excretion of GHB [48,49].

As there are no effective specific treatments for SSADH deficiency in people, treatment is currently aimed at managing the clinical signs of seizures and neurobehavioral disturbances [34]. Broad-spectrum anticonvulsants are generally utilized, avoiding valproate which inhibits SSADH which may worsen GHB accumulation and clinical signs [50]. However, there are many therapeutic options under investigation which include pharmacological (e.g., targeting neurotransmitter receptors), enzyme-replacement therapy, gene therapy and treatment with pharmacological chaperones [51,52].

In animals, SSADH has been produced in knockout mice; clinical signs are progressive ataxia, failure to thrive, seizure and death at a young age [53,54], and thus they are utilized as a model for the severe and poor survival phenotype in humans. There are no reports of the MR imaging features of SSADH-knockout mice [51] and, on pathology, there are no reported abnormalities on routine hematoxylin and eosin staining, but detailed neuropathological examinations have not been performed [55]. There is a single case report of a dog with suspected SSADH deficiency, which had a progressive encephalopathy with profound and persistent lactic acidosis, elevated urine GHB and a 30% reduced activity of SSADH measured in cultured lymphoblasts compared to normal dogs. Intracranial MR imaging of the brain was not performed; on histopathology of the brain, there was a spongiform change in the cerebral cortex [56].

Saluki dogs have a more severe phenotype of SSADHD than people, but with very similar clinical signs of seizures, abnormal behavior, abnormalities of sleep and multifocal brain disease on neurological examination. Like people [42], affected Salukis have multifocal abnormalities in the brain on MR imaging, where affected dogs have identical bilaterally symmetrical T2-weighted hyperintensities in the same anatomical areas as people such as the basal nuclei, deep cerebellar nuclei, and brain stem. In people with SSADH deficiency, these abnormalities have considerable consistency, and are almost universal. However, in some people, there are reports of non-specific hyperintensities in subcortical white matter and the substantia nigra in the brainstem, as well as cerebellar atrophy [57,58]. These abnormalities are not considered specific for SSADHD, but considered to be imaging characteristics of cytotoxic edema, secondary to oxidative stress from the underlying SSADHD [43]. Differently to what is noted in people, dogs have hyperintensity of the deep cortical laminae of the grey matter of the cerebral cortex and atrophy of the cerebral cortex. On histopathology in dogs with SSADH deficiency, there is bilaterally symmetric multifocal spongiform change in the brainstem, deep cerebellar nuclei, but also severe in the thalamic nuclei and deep cortical grey matter. The brain lesions in affected Salukis on MRI (two dogs) and pathology (three dogs (same two dogs that had MRI plus one other) were identical. There is only one case report of the pathology of SSADHD in a person where the histopathology was not described and thus the comparative pathology between dogs and people is not possible [42].

Unlike in people with SSADHD, where GHB is elevated in the urine and therefore is an excellent biomarker, GHB is not elevated in the urine of affected dogs. GHB is, however, elevated in the serum, CSF and brain tissue of affected dogs. Since GHB is extensively reabsorbed from the proximal tubules in the kidney [59], it is possible that species differences in handling GHB resulted in extensive reabsorption of GHB and in a normal level of GHB in the affected dog’s urine. It is also plausible that since urine GHB was evaluated in only two affected dogs, that evaluation of additional SSADHD-affected dogs may have yielded elevated levels of urine GHB. There may also have been loss of GHB in the samples during shipping to the lab or storage, as GHB is a volatile compound [40] and a reduction in GHB levels is reported with storage, which varied between 10% loss at just 3 days and in excess of 20% after 4 weeks of storage [60]. If the activity of the D-2-hydroxyglutarate dehydrogenase in liver (non-cofactor enzyme converting GHB to SSA and ketoglutarate to D-2-hydroxyglutarate) was very active in the liver of Salukis, this could be a plausible explanation for why GHB was not elevated in the urine; however, D-2-HG was not measured in affected or control dogs. Otherwise, aside from serum SSA that we were not able to measure in dogs with SSADHD, dogs are similar to human patients in regards to elevated body fluid levels of SSA, DHHA and GHB, and zero to low levels of brain SSA activity.

## 5. Conclusions

Saluki dogs with SSADHD have a disease phenotype resembling SSADHD in people, although it appears to be more severe clinically. On the other hand, it appears to be less severe than the phenotype described in in knockout mice, which is lethal [58]. Furthermore, GHB may not be an acceptable biomarker for the disease in dogs; alternatively, urine SSA or GHB in serum may be more appropriate biomarkers. Compared to mice models of human disease, dog models have naturally occurring disease, are more similar to humans in regards to size, and have more longevity than mice. Dogs are proven and valuable models of human disease, particularly in the field of lysosomal storage diseases [6]. This first *ALDH5A1*-related large animal model for SSADHD may provide an opportunity for evaluation of potential therapeutics for this rare orphan disease in people. Dogs may be a more appropriate disease model than the murine SSADH model, as dogs appear to have a more similar disease phenotype and similar MR imaging features to people. The identification of the pathogenic *ALDH5A1* variant will allow the screening of carriers to avoid producing further affected puppies and thereby contribute to maintaining breed health.

## Figures and Tables

**Figure 1 genes-11-01033-f001:**
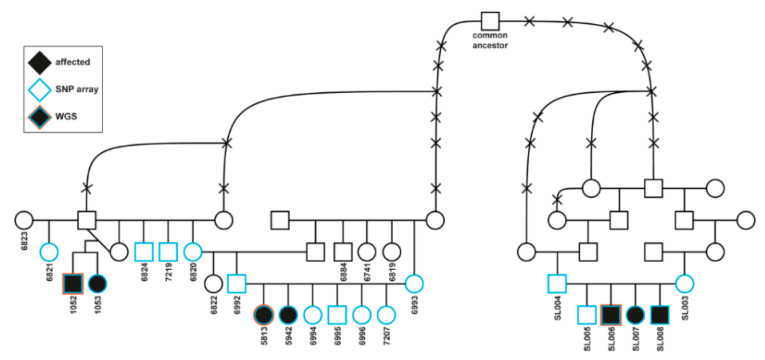
Pedigree of seven succinic semialdehyde dehydrogenase deficiency (SSADHD)-affected Saluki dogs. Females are depicted by circles and males by squares. Black fill indicates affected puppies. Numbers indicate 25 dogs from which samples were available. The blue contour indicates animals that were genotyped on SNP array, and the red contour the three affected dogs selected for the WGS. Note that the two litters on the left were seen in the USA and the third litter on the right in Germany. A common male ancestor illustrates the genealogical relatedness.

**Figure 2 genes-11-01033-f002:**
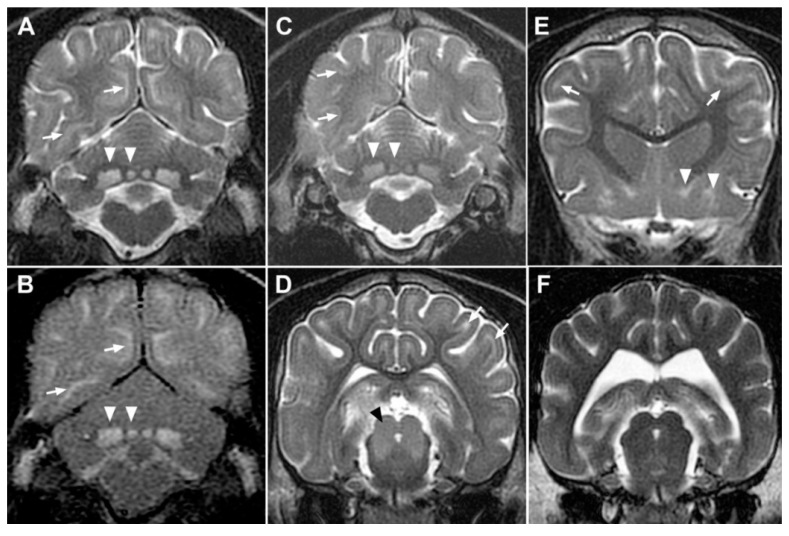
MRI abnormalities in SSADHD-affected Saluki Dogs. Transverse T2-weighted (**A**,**C**–**F**) and FLAIR (**B**) MR images at the level of the cerebellum (**A**–**C**), midbrain (**D**,**F**) and corpus striatum (**E**) demonstrating symmetrical involvement of predominantly grey matter structures. Images from dog 1 (**A**,**B**,**D**,**E**), dog 2 (**C**) and an unaffected littermate (**F**). Consistent bilateral symmetrical T2 hyperintensity of the deep cerebellar nuclei ((**A**–**C**); white arrowheads) is the most prominent finding. Similar bilaterally symmetrical hyperintensity is seen involving the tectum and dorsal tegmentum ((**D**); black arrowhead) and endopeduncular (medial) and lentiform nuclei ((**E**); white arrowheads) but not present in unaffected dog images (**F**). Sulci are prominent (**D**) compared to an unaffected age matched control (**F**), consistent with atrophy of cortical grey matter. Hyperintensity of deep cortical grey mater laminae is evident on T2-weighted and FLAIR images at all levels (white arrows) is not present on unaffected dog MR images (**F**).

**Figure 3 genes-11-01033-f003:**
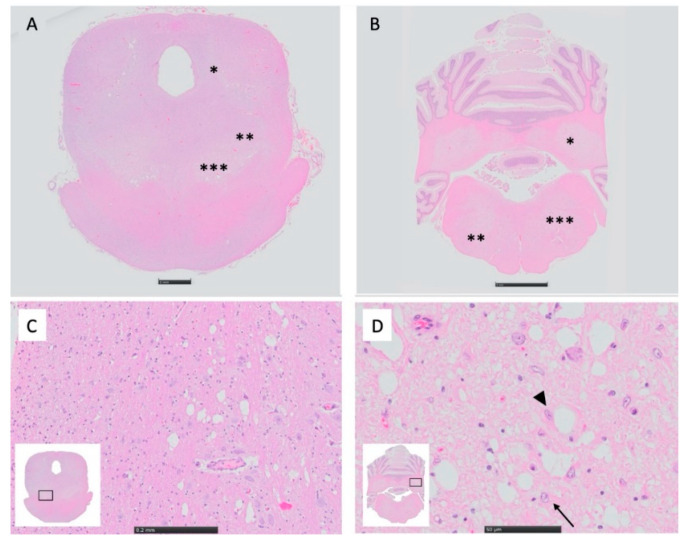
Histopathology of mesencephalon and brainstem from dog 1. (**A**) Bilaterally, the mesencephalic nuclei of cranial nerve V (*), the red nuclei (**), and the substantia nigra (***) exhibit decreased staining intensity (H&E). (**B**) Bilaterally, the deep cerebellar nuclei (*), the dorsal nuclei of the trapezoid body (**), and the reticular formations (***) exhibit decreased staining intensity (H&E). (**C**) Higher magnification of (**A**), inset. The substantia nigra shows prominent vacuolation of affected neurons. (**D**) Higher magnification of (**B**), inset. The interposital nucleus shows reactive astrocytes (arrow), some of which also contain prominent cytoplasmic vacuolation (arrowhead).

**Figure 4 genes-11-01033-f004:**
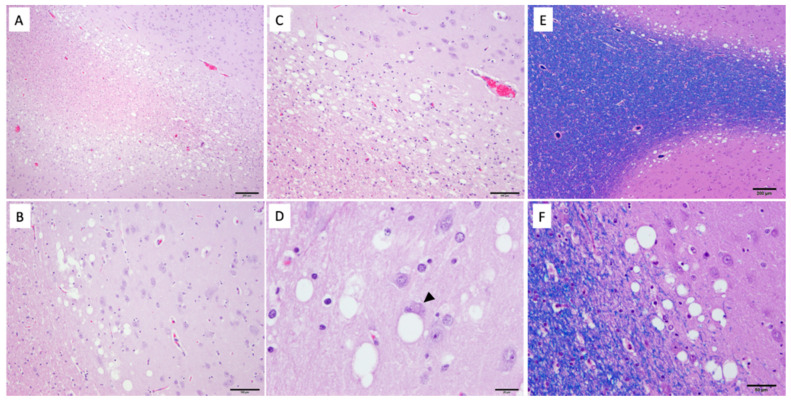
Histopathology of forebrain from dog 1. (**A**) Within the frontal cortex, the grey matter is predominantly affected by spongiotic change, with gliosis. (**B**) The spongiosis within the forebrain is most severe in the deep laminar cortex. (**C**) The caudate nucleus is also affected at the grey–white matter junction. (**D**) Affected neurons are characterized by enlarged, vacuolated cytoplasm with a peripheralized nucleus (arrowhead). Luxol fast blue (LFB) staining highlights the deep cortical nature of the vacuolation (**E**) within the cerebrum. (**F**) Vacuolation is discrete and often displaces cellular nuclei LFB staining.

**Figure 5 genes-11-01033-f005:**
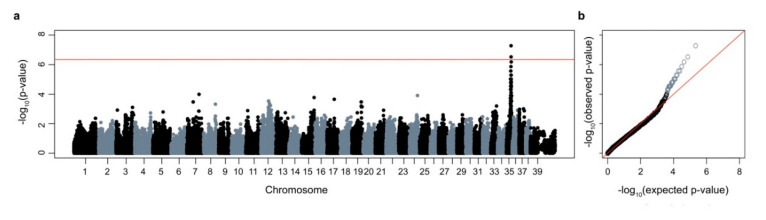
GWAS for SSADHD-affected Saluki dogs. (**a**) Manhattan plot showing –log10 of the raw p-values for each genotyped SNP by chromosome (*x*-axis). Genomic inflation was 1.25. Line denotes genome-wide significance based on Bonferroni-corrected p-values. (**b**) Q–Q plot of samples used in GWAS showing the −log10 of the expected versus the observed p-values. The SNPs on CFA35 are shown in light grey.

**Figure 6 genes-11-01033-f006:**
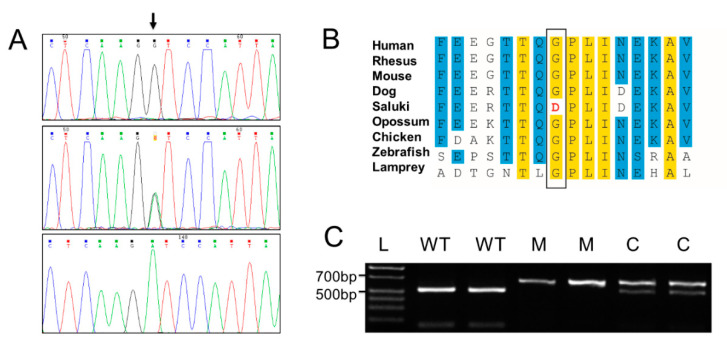
SSADHD-associated *ALDH5A1* missense variant in Saluki dogs. (**A**) The electropherograms from a normal dog (top panel), a heterozygous dog (middle panel) and a dog homozygous for the variant in *ALDH5A1* indicated by an arrow. (**B**) The amino acid alignment around the missense variant in ALDH5A1 (XP_013966074.2: p.(Gly288Asp)). Yellow boxes indicate 100% conservation across the species listed to the left and blue boxes indicate 75% conservation. The variant amino acid residue is boxed and the variant allele detected in affected Salukis is shown in red. (**C**) The PCR-RFLP genotyping assay for the *ALDH5A1* missense variant is shown. After PCR amplification, the products were digested with Sau96I. L is the DNA ladder, WT stands for wild type (542 bp), C for carrier and M for mutant (702 bp).

**Figure 7 genes-11-01033-f007:**
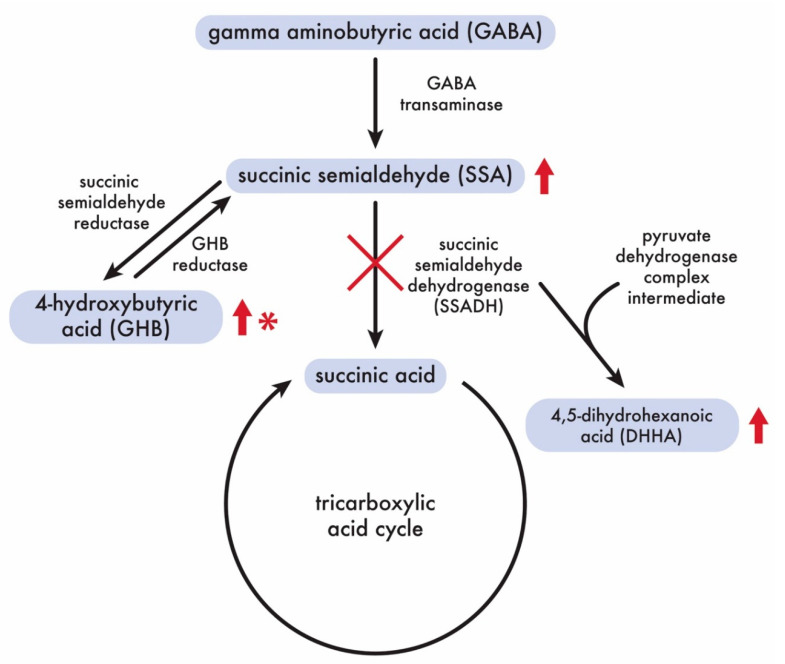
GABA catabolism pathway. In Saluki dogs with SSADH deficiency, levels of SSA and DHHA are elevated in urine, serum, CSF and brain, and GHB is elevated in serum, CSF and brain (red arrows) as in people with SSADH deficiency. Unlike in people, where GHB is elevated in urine (red arrow), the level of GHB in urine (red *) in Saluki dogs with SSADH deficiency is normal.

**Table 1 genes-11-01033-t001:** Clinical information for seven SSADHD-affected Saluki dogs. (OU = oculus uterque (both eyes)).

Dog No.	Sample ID	Country	Sex	Age of Onset	Clinical Signs	Neurological Examination	Outcome
1	5813	USA	F	10 weeks	Generalized epileptic seizures, episodes of vocalization, abnormal behavior, generalized ataxia with thoracic limb hypermetria	Mild generalized ataxia with thoracic limb hypermetria. Delayed proprioceptive positioning present in all 4 limbs	Treated with anticonvulsants (phenobarbital), euthanized at 32 weeks of age
2	4942	USA	F	10 weeks	Generalized epileptic seizures, episodes of vocalization, abnormal behavior, generalized ataxia with thoracic limb hypermetria	Not done	Treated with anticonvulsants (phenobarbital), euthanized at 39 weeks of age
3	1053	USA	F	6 weeks	Focal epileptic seizures, episodes of vocalization, normal between episodes Unable to arouse when sleeping	Absent menace response OU	Treated with anticonvulsants (phenobarbital), euthanized at 17 weeks of age
4	1052	USA	M	6 weeks	Generalized and focal epileptic seizures, episodes of vocalization, normal between episodes. Unable to arouse when sleeping	Absent menace response OU	Treated with anticonvulsants (phenobarbital), euthanized at 17 weeks of age
5	SL006	Germany	M	9 weeks	Focal epileptic seizures, deep sleep	Thoracic limb hypermetria, mild ataxia, reduced proprioceptive positioning, absent menace	Treated with levetiracetam, euthanized at 4 months of age
6	SL008	Germany	M	9 weeks	Focal epileptic seizures, episodes of vocalization	Not done	Euthanized at unknown age
7	SL007	Germany	F	9 weeks	Focal epileptic seizures	Not done	Euthanized at unknown age

**Table 2 genes-11-01033-t002:** Specific quantitative organic acids in urine, serum, CSF, and brain tissue in affected and control dogs (nd = not done).

Dog Number	Urine SSA, mmol/mol Creatinine	Urine GHB, mmol/mol Creatinine	Urine DHHA mmol/mol Creatinine	Serum GHB, µmol/L	Serum DHHA, µmol/L	CSF SSA, µmol/L	CSF GHB, µmol/L	CSF DHHA, µmol/L	Brain GHB, nmol/mg Brain	Brain DHHA, nmol/mg Brain	Brain SSA, nmol/mg Brain	Brain SSA Activity, pmol/min/mg Protein
1	9.23	1.06	5.85	6.59	0.45	69	>1500	43.3	2.43	0.22	0.23	10
2	nd	nd	nd	nd	nd	nd	nd	nd	2.93	0.28	0.18	0
5	38.7	nd	11.8	nd	0.41	nd	nd	nd	nd	nd	nd	nd
6	30.9	0.67	10.4	nd	0.61	nd	nd	nd	nd	nd	nd	nd
7	nd	nd	nd	nd	0.56	nd	nd	nd	nd	nd	nd	nd
Number of affected dogs	3	2	3	1	4	1	1	1	2	nd	2	2
Number of control dogs	4	4	4	4	4	2	3	3	4	nd	nd	4
median affected	30.9	0.87	10.4	n/a	0.51	n/a	n/a	n/a	2.68	0.25	0.21	5
range affected	9.23–38.7	0.67–1.06	5.85–11.8	n/a	0.41–0.61	n/a	n/a	n/a	2.43–2.93	0.22–0.28	0.18–0.23	0–10
median control	0.86	0.82	0.29	0.38	0.08	0.24	0.31	0.11	0.03	0	0.11	5587
range control	0.64–0.9	0.29–2.04	0.18–0.65	0.28–0.59	0.07–0.1	0.02–0.46	0.23–0.8	0.1–0.2	0–0.05	0–0	0.06–0.14	4214–5942

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
