# Peer review of "A Missense Variant in ALDH5A1 Associated with Canine Succinic Semialdehyde Dehydrogenase Deficiency (SSADHD) in the Saluki Dog"

_genes, 2020, doi:10.3390/genes11091033_

Round 1

Reviewer 1 Report

In the article "A missense variant in ALDH5A1 associated with canine succinic semialdehyde dehydrogenase deficiency (SSADHD) in the Saluki dog" Vernau et al. investigate seven Saluki puppies for neurological symptoms. Using genome-wide association study and whole-genome sequencing they find homozygous variants in the ALDH5A1 gene to be causative for the clinical changes and demonstrate biochemical alterations associated with the changes in the affected metabolic pathway. Additionally, the authors reveal distinct structural changes in the affected dog's brain found on MRI and discuss assocaited histopathological observations and conclude that a condition formerly termed "Status spongiosi in Saluki dogs" (SSSD) can be explained by genetic variants in ALDH5A1.

The article is extremely well written and explains the scientific approach, the methods and the conclusions drawn from them in an impressive and at the same time conclusive way.

Some minor remarks:
I have a note on Figure 1, in which the description mentions 24 numbered dogs, for which samples were available. However, I can count 25 dogs, under circumstances this may simply be a typo, please check.

In the methods section, it is mentioned that the dogs were examined in the USA and Germany, and that the biological material was obtained and analyzed in these two countries. While for the USA there is an approval of the regulating authority, the approval for the German study subjects seems to have been granted in the canton of Bern in Switzerland. If this is legally correct (I am not familiar with the veterinary regulations), my comment can be ignored.

In the discussion part, I would like the authors to go into more detail about the consistency of the MRI findings of the dogs. We know from patients that the structural changes are extremely heterogeneous and do not carry much diagnostic value. It seems that a much stronger structural phenotype is found in the dog model. Is this due to the relatively close relationship of the dogs? What do the authors think about this?

In addition, I would be interested as a Reader, in how far dogs are already used as a model for other neurometabolic diseases. One short sentence would be enough.

Author Response

Thank-you very much for your detailed review and constructive comments about our manuscript.

  1. I have a note on Figure 1, in which the description mentions 24 numbered dogs, for which samples were available. However, I can count 25 dogs, under circumstances this may simply be a typo, please check.

Thank-you you are correct, 25 dogs, we corrected the figure to 24 dogs.

  1. In the methods section, it is mentioned that the dogs were examined in the USA and Germany, and that the biological material was obtained and analyzed in these two countries. While for the USA there is an approval of the regulating authority, the approval for the German study subjects seems to have been granted in the canton of Bern in Switzerland. If this is legally correct (I am not familiar with the veterinary regulations), my comment can be ignored.

Correct, affected dogs were examined in the USA and Germany. Affected dogs were clinical patients (and thus institutional approval is not required for diagnostic testing) and samples were drawn in Germany as part of their routine diagnostic testing by a licensed veterinarian. Institutional approval was sought when samples from the German dogs were submitted to the laboratory in Bern.

  1. In the discussion part, I would like the authors to go into more detail about the consistency of the MRI findings of the dogs. We know from patients that the structural changes are extremely heterogeneous and do not carry much diagnostic value. It seems that a much stronger structural phenotype is found in the dog model. Is this due to the relatively close relationship of the dogs? What do the authors think about this?

We updated the paragraph from Line 524-537

  1. In addition, I would be interested as a Reader, in how far dogs are already used as a model for other neurometabolic diseases. One short sentence would be enough.

We added this sentence, line 562 to 565:

Compared to mice models of human disease, dog models have naturally occurring disease, are more similar to humans in regards to size, and have more longevity than mice. Dogs are proven and valuable models of human disease, particularly in the field of lysosomal storage diseases.

Reviewer 2 Report

Karen M. Vernau et al. conducted an original and interesting study reporting seven SSADHD-affected Saluki dogs.

Comprehensive evaluations including MRI, necropsy and GWAS followed by whole-genome sequence analysis of three affected puppies. Metabolic and enzyme activity testing were performed on serum, urine, CSF and brain tissue.

The applied methodology and statistical analysis were correct, and the results demonstrated a homozygous missense variant in ALDH5A1 gene (XM_014110599.2: c.866G>A; 45 XP_013966074.2: p.(Gly288Asp). Levels of SSA and DHHA are elevated in urine, serum, CSF and brain. Levels of GHB is elevated in serum, CSF and brain. Unlike people the level of GHB in urine is normal.

The authors conclude that affected saluki dogs had striking similarities to SSADH deficiency in humans and provides a unique translational large animal model for the development of novel therapeutic strategies.

The discussion is well thought out and up to date.

The resulting paper is well designed and written, easy to read and understand.

Author Response

Thank-you Reviewer 2 for your review of our manuscript. 

Sincerely, 

Karen Vernau